# Multiple Myeloma Derived Extracellular Vesicle Uptake by Monocyte Cells Stimulates IL-6 and MMP-9 Secretion and Promotes Cancer Cell Migration and Proliferation

**DOI:** 10.3390/cancers16051011

**Published:** 2024-02-29

**Authors:** Rebecca Sheridan, Kieran Brennan, Despina Bazou, Peter O’Gorman, David Matallanas, Margaret M. Mc Gee

**Affiliations:** 1School of Biomolecular and Biomedical Science, University College Dublin, Belfield, D04 V1W8 Dublin, Irelandk.brennan@ucd.ie (K.B.); 2Department of Haematology, Mater Misericordiae University Hospital, D07 R2WY Dublin, Ireland; despina.bazou@ucd.ie (D.B.);; 3School of Medicine, University College Dublin, Belfield, D04 V1W8 Dublin, Ireland; 4Systems Biology Ireland, School of Medicine, University College Dublin, Belfield, D04 V1W8 Dublin, Ireland; david.gomez@ucd.ie; 5Conway Institute of Biomolecular and Biomedical Research, University College Dublin, Belfield, D04 V1W8 Dublin, Ireland

**Keywords:** cancer, extracellular vesicles, multiple myeloma, bone marrow microenvironment, metastasis, spliceosome

## Abstract

**Simple Summary:**

Multiple myeloma is an incurable blood cancer that arises due to the uncontrolled growth of plasma cells in the bone marrow. The progression of multiple myeloma is dependent on communication between the plasma cells and other cells within the bone marrow, and extracellular vesicles (EVs) are implicated in the cell communication. The aim of this study was to investigate the role of EVs secreted by multiple myeloma cells on monocyte cells. We revealed that multiple myeloma cell-derived EVs are taken up by monocyte cells. EV uptake by monocytes led to increased secretion of pro-inflammatory factors, which in turn created an environment that enhanced myeloma cell migration and growth that is associated with cancer progression. Investigation into the EV contents revealed potential mechanisms by which multiple myeloma EVs promote cancer progression.

**Abstract:**

Multiple Myeloma (MM) is an incurable haematological malignancy caused by uncontrolled growth of plasma cells. MM pathogenesis is attributed to crosstalk between plasma cells and the bone marrow microenvironment, where extracellular vesicles (EVs) play a role. In this study, EVs secreted from a panel of MM cell lines were isolated from conditioned media by ultracentrifugation and fluorescently stained EVs were co-cultured with THP-1 monocyte cells. MM EVs from three cell lines displayed a differential yet dose-dependent uptake by THP-1 cells, with H929 EVs displaying the greatest EV uptake compared to MM.1s and U266 EVs suggesting that uptake efficiency is dependent on the cell line of origin. Furthermore, MM EVs increased the secretion of MMP-9 and IL-6 from monocytes, with H929 EVs inducing the greatest effect, consistent with the greatest uptake efficiency. Moreover, monocyte-conditioned media collected following H929 EV uptake significantly increased the migration and proliferation of MM cells. Finally, EV proteome analysis revealed differential cargo enrichment that correlates with disease progression including a significant enrichment of spliceosome-related proteins in H929 EVs compared to the U266 and MM.1s EVs. Overall, this study demonstrates that MM-derived EVs modulate monocyte function to promote tumour growth and metastasis and reveals possible molecular mechanisms involved.

## 1. Introduction

Multiple myeloma (MM) is a haematological malignancy of the B-cell lineage, characterised by the monoclonal expansion of plasma cells in the blood and bone marrow (BM). It is the second most common blood cancer and despite advancements in MM treatment in the last decade that have led to an increase in the 10-year survival rates, MM remains incurable [1,2]. The progression of MM occurs initially with the development of monoclonal gammopathy of undetermined significance (MGUS), later developing into smouldering MM, and finally into symptomatic myeloma [1]. During disease development, malignant plasma cells interact with the bone marrow microenvironment (BMM) cells to establish a tumour niche that is beneficial to their growth and survival [1], resulting in the development of an immunosuppressive tumour microenvironment (TME). Hence, an increased understanding of MM progression and metastasis is vital to reduce disease progression and mortality.

The TME, a critical regulator in the progression of MM, consists of various cells within the bone marrow as well as their secreted soluble factors and signals. The microenvironment undergoes constant change based on the cells present and the signals released and exchanged. Such alterations produce a microenvironment that is receptive to cancer dissemination. Furthermore, modifications within the metastatic niche can promote MM proliferation, metastasis, and disease progression [3]. Monocytes are key regulators of the BMM and an integral part of the immune system. Modulation of monocytes by cancer cells is a key step for angiogenesis, proliferation, metastasis, and disease progression [4,5]. Monocytes are trafficked through the bloodstream to tissue at a steady rate; during inflammation, this rate is increased and monocytes can be transformed into different tissue macrophages and dendritic cells [6]. Monocytes are categorised into classical (pro-inflammatory) and non-classical (anti-inflammatory) forms that can stimulate and inhibit tumour growth depending on their phenotype and other TME factors [7,8,9]. It is believed that anti-inflammatory, non-classical monocytes are derived from classical monocytes in the bloodstream, suggesting that modifications in their transcriptome and proteome, likely driven by external signals such, as cytokines, chemokines and EVs, regulate phenotype switching [10].

MM cells are reliant on the pro-inflammatory properties of the tumour microenvironment for the release of key signals involved in their growth, survival, migration, and proliferation. Two key regulators in MM survival and progression are interleukin-6 (IL-6) and matrix metallopeptidase 9 (MMP-9). Although MM cells can produce IL-6, the major source is from the TME [11,12]. It has been shown that co-culture of MM BM-derived cells with MM cells, compared to normal BM cells, results in increased expression and secretion of IL-6 by MM-derived BM cells [13]. MMP-9 mediates the degradation of extracellular matrix proteins and promotes tissue remodelling and extracellular matrix remodelling mediated by bone marrow stromal cells, supporting the invasion of endothelial cells that form neoangiogenic blood vessels, and infiltration and growth of MM cells within the BMM [14]. Considering the essential roles of IL-6 and MMP-9 in MM survival and progression, it is important to understand how MM cells modulate their expression and release within the BMM.

EVs are membrane-bound nanoparticles that consist of lipid bilayer containing cargo representative of their cell of origin such as transmembrane proteins, enclosed cytosolic proteins and RNA [15]. EVs can be characterised by their size and mechanism of release: EVs released from budding cells vary in size (100–1000 nm) and are known as microvesicles, ectosomes and microparticles, whereas EVs released from multi-vesicular bodies (MVB) are known as exosomes and have a size of 30–150 nm, and the majority of apoptotic bodies fall into the range of 500–5000 nm [15,16]. EVs are key mediators of cell communication, and they are secreted by all cells within the body, however, malignant cells secrete higher EV numbers [17]. EV communication within the local TME and distant bone marrow environment is believed to play a pivotal role in MM disease progression and the development of drug resistance [18,19]. For example, EVs remodel the BMM to promote bone lesion formation, angiogenesis at pre-metastatic BM sites and immune cell evasion [20].

In this study, we investigate the role of MM-derived EVs in the modulation of monocyte function towards a pro-metastatic form. We reveal that MM EVs alter monocyte function by inducing the secretion of IL-6 and MMP-9 and promoting MM cell migration and proliferation. Furthermore, through EV proteomics, we identify a unique EV cargo and associated molecular pathways that may promote the pro-metastatic phenotype.

## 2. Materials

The primary antibodies: anti-CD147 (1:1000, ThermoFisher, Waltham, MA, USA, MA1–19,201), anti-HSP70 (1:3000, Santa Cruz, CA, USA, sc-66,048), anti-TSG101 (1:1000 dilution, Abcam, Cambridge, UK, ab125011), anti-HSP90B1 (1:1000 dilution, Cusabio, TX, USA, CSBPA10887A0Rb), anti-CD9 (1:1000, ThermoFisher, MA5-31980), anti-TFAM (1:1000, Cell Signalling, Beverly, MA, USA, #8076), anti-NRAS (1:500, Santa Cruz, sc-519), anti-ZAP70 (1:1000, Cell signalling, #3165), anti-GAPDH (1:1000, Santa Cruz, sc-32233). The secondary antibodies: anti-Rabbit IgG-DyLight 800 (1:3000 dilution, ThermoFisher, SA5-35571), anti-Mouse IgG-DyLight 680 (1:3000 dilution, ThermoFisher, 35,519) Reagents: Trypsin (Pierce Trypsin Protease, MS Grade, ThermoFisher, 90,057), Propidium Iodide (ThermoFisher, P1304MP), Far Red CellTrace (ThermoFisher, C34564).

### 2.1. Cell Culture

Myeloma cell lines H929 (ATCC: CRL-9068) and MM.1s (ATCC: CRL-2974), and human monocytic cell line THP-1 (ATCC: TIB-202) were cultured in RPMI 1640 medium (Gibco, Waltham, MA, USA) with 10% foetal bovine serum (FBS) (Gibco) and 1% Penicillin-Streptomycin (Gibco). The myeloma cell line U266 (ATCC: TIB-196) was cultured in RPMI 1640 medium (Gibco) with 15% foetal bovine serum (FBS) (Gibco) and 1% Penicillin-Streptomycin (Gibco). RPMI 1640 medium with 1% Penicillin-Streptomycin only was used as a serum-free medium. Cells were cultured at 37 °C in a 5% CO_2_ humidified incubator and passaged every 2–3 days.

### 2.2. Preparation of Conditioned Media for EV Isolation

H929, U266 and MM.1s cells were grown in complete media (RPMI 1640 (10–15% FBS, 1% Pen/Strep) in a T175 cell culture flasks at 37 °C under standard conditions. Cell viability was checked by trypan blue exclusion to ensure it was >90%. Cells were collected by centrifugation at 450× *g* for 3 min. Cells (1 × 10^6^ cells/mL) were cultured in serum-free media (RPMI 1640 (1% pen/strep)) in 145/20 mm cell culture dishes for 48 h. The conditioned media from the cultures was collected and used for EV isolation. Cells were removed by centrifugation at 450× *g* for 3 min, and cell debris was removed by centrifugation at 2000× *g* for 20 min. EV isolation was carried out immediately following conditioned media collection.

### 2.3. Ultracentrifugation and Iodixonol Density Gradient for EV Isolation

MM-derived cell culture conditioned media was centrifuged at 120,000× *g*_Avg_ (SW32Ti, Beckman Coulter (Miami, FL, USA), brake = 9) for 2 h 20 min at 20 °C (centrifugation durations determined using a “50 nm cut-off size” as per [21]). The pellet was washed in 1 mL PBS and centrifuged at 120,000× *g*_Avg_ for 35 min (MLA-130, Beckman Coulter, brake = 9).

The EV-enriched pellet underwent floatation density gradient for a high-purity EV isolation. The EV-enriched pellet was resuspended in 100 µL of PBS. EV-enriched solution was mixed with 335 µL of 54% iodixanol-PBS (IPBS). The volume was made up to 1 mL with 1.2 g/mL IPBS. Samples were centrifuged at 120,000× *g*_Avg_ for 15 h at 4 °C. The EV layer that formed in the 1.2 g/mL fraction was transferred to a new tube and diluted to less than 1.03 g/mL in PBS. The samples were then centrifuged for 35 min at 20 °C (MLA-130, Beckman Coulter, brake = 9). The resulting high-purity EV-enriched pellet was resuspended in 100 µL PBS and stored at −80 °C.

### 2.4. Western Blot

Cell or EV proteins were extracted in RIPA lysis buffer supplemented with protease cocktail inhibitor (Abcam, Ab201113). The lysed samples were incubated on ice for 10 min and centrifuged at 20,000× *g* for 30 min, 4 °C. Cell and EV lysate protein concentration was determined using bicinchoninic acid (BCA) protein assay kit (Thermo Fisher Scientific), carried out according to manufacturer’s protocol. Protein samples were diluted to a determined concentration in PBS and samples were reduced using 4× Laemmli buffer followed by heating at 95 °C for 5 min. Protein separation was carried out on a 5−12% SDS-PAGE gel using a Bio-Rad MiniProtean II gel system. Separated proteins were transferred to 0.45 μm PVDF membrane using a mini-Protean II blotting system at 110 V constant voltage for 80 min. Protein membranes were blocked using 5% bovine serum albumin (BSA) in Tris-buffer saline—tween20 (0.1%) (TBS-T) for 1 h at RT. The membranes were probed with primary antibodies in 5% BSA (TBS-T) at 4 °C overnight. The membranes were washed three times with TBS-T and then incubated with IRDye800-conjugated goat anti-rabbit IgG or IRDye680-conjugated goat anti-mouse IgG secondary antibodies diluted 1:3000 in 5% BSA blocking buffer for 1 h at RT. Next, the blots were washed six times for 5 min with TBS-Tween and TBS alternating washes. Blots were visualised using Odyssey Infrared Imaging System (LI-COR Biosciences, Lincoln, NE, USA) with both 700- and 800-nm channels.

### 2.5. Nanoparticle Tracking Analysis

The size and particle concentration of EV preparations were measured by a NanoSight NS300 system (Malvern Technologies, Middlesex, UK) combined with a 488 nm laser and a high-sensitivity scientific CMOS camera. Samples were diluted in the range of 1:100–1:200 in particle-free PBS (Gibco) to a detectable concentration of 30–60 particles per frame. Samples were analysed under constant flow conditions (flow rate = 50) at 25 °C and 15 captures for 60 s/capture were analysed using NTA3.2 software with a detection threshold of 5 and a bin size of 2.

### 2.6. Transmission Electron Microscopy

TEM grids (formvar carbon coated 200 mesh copper grid) were incubated with 10 μL of isolated EV samples for 60 min. The grids were rinsed in PBS 3 times for 2 min each and dried with filter paper. The samples were fixed by adding 2.5% glutaraldehyde for 10 min followed by washing the grids in distilled water 5 times for 2 min each. 2% uranyl acetate was added onto the grids and incubated for 15 min at room temperature. The grids were rinsed quickly with ice-cold 1.8% methylcellulose and 0.4% uranyl acetate (MC/UA). The samples were embedded by adding MC/UA for 10 min on ice. The grids were air-dried at room temperature and examined by TEM (FEI Tecnai^TM^ 120 kV transmission electron microscope).

### 2.7. Cell Viability Quantification

#### 2.7.1. Trypan Blue

Cell viability for MM cell lines during EV preparation was determined by Trypan blue (Gibco) exclusion. Cells were washed twice in PBS. Cells were diluted 1:4 with Trypan blue and viable cells were counted using a haemocytometer.

#### 2.7.2. Flow Cytometry

For THP-1 uptake studies, cells were washed in PBS twice at 450× *g* for 3 min. The cells were then resuspended in 100 µL of PBS and stained with 10 µg/mL propidium iodide (PI). Cells were incubated on ice in the dark for 1 min prior to analysis using the Cytoflex LX flow cytometer (Beckman Coulter, Brea, CA, USA). PI was detected using the Y585-PE channel. The stop count was set at 10,000 total events. The percentage of cell viability was determined by measuring the negative population (established with the unstained control).

### 2.8. EV Quantification

Flow cytometry analysis was performed on the Beckman Coulter CytoFLEX LX and S Flow Cytometer. Daily calibration of flow cytometer CytoFLEX Daily QC beads as per manufacturer specifications, followed by Apogee Mix Silica (Si) and polystyrene (PS) beads (Apogee Flow Systems Ltd., Middlesex, UK) were used in sizes of PS 80 nm, PS 110 nm, Si 180 nm, Si 240 nm, Si 300 nm, PS 500 nm, Si 590 nm, Si 880 nm and Si 1300 nm. Extensive cleaning cycles were carried out using both daily clean and deep clean cycles. Gaintration was carried out to establish fluorescent gain values. The scatter gain (blue, red, and violet laser) and threshold were established using the Apogee Mix beads (VSSC gain = 300; VSSCsingle bondH threshold = 5500). Events were gated to remove EV aggregates; a further gate was added to contain the 80 nm to 500 nm bead population. The gating strategy further outlined in Appendix A. To avoid swarm effects each was serially diluted from 1:2 to 1:500 to achieve an event count of 6000 events/s and measured with a flow rate of 10 μL/min.

### 2.9. EV Uptake of THP-1 Cells and Conditioned Media Collection

Prior to the density gradient step in the EV isolation protocol, far red CellTrace (frCellTrace) (Invitrogen, Waltham, MA, USA, C34564) was used to label the EVs. The EV-enriched pellet was resuspended in 100 µL of PBS. frCellTrace was added at a concentration of 3.34 µM, and incubated in the dark at RT for 10 min. The EV samples were then isolated following the density gradient protocol. EV labelling is carried out prior to density gradient separation to allow for the removal of excess dye in this step. The EV samples were quantified using flow cytometry as described above. THP-1 cells (2.5 × 10^5^ cells/mL) were co-cultured with a range of EV concentrations (25–400 EVs/cell) for 24 h at 37 °C. The cells were collected by centrifugation at 450× *g* for 3 min. The supernatant comprising of the THP-1 conditioned media was additionally centrifuged at 10,000× *g* for 10 min to remove excess cell debris and transferred to a new tube stored at −20 °C. for use in downstream assays and EV uptake by cells was determined by detection of APC fluorescence using the Cytoflex LX (Beckman Coulter).

### 2.10. Confocal Microscopy of EV Uptake

Following 24 h co-incubation with MM EVs, THP-1 cells were incubated in PBS with 5 ng/µL Cell Mask Orange plasma membrane stain (Invitrogen, C10045) for 10 min at 37 °C. Cells were centrifuged onto a slide using a cytospin at 700 RPM for 2 min. The cells were fixed in 4% paraformaldehyde at room temperature for 10 min and washed twice with PBS. Fixed cells were incubated with 300 nM 4′,6-diamidino-2-phenylindole (DAPI, Merck, NJ, USA, D9542) for 1 min and washed twice with PBS before using MOWIOL (Sigma-Aldrich, Saint Louis, MO, USA) mounting media to fix cover slide. Zeiss LSM 800 Airy confocal microscopy was used to image cells and Zen Blue was used for image analysis.

### 2.11. ELISA Assay

Supernatants isolated from THP-1 cells were used to quantify MMP-9 using a human MMP-9 DuoSet ELISA (R&D Systems, Minneapolis, MN, USA, DY911) according to the manufacturer’s instructions. IL-6 levels were quantified using human IL-6 specific ELISA kits (Human IL-6 SuperSet ELISA Kit, ELISAGenie, Dublin, Ireland, HUDC0132) according to the manufacturer’s protocol.

### 2.12. Migration Assay

THP-1 conditioned media, isolated following 24 h co-incubation with H929-derived EVs, PBS and untreated negative controls, was tested for its chemoattraction activity towards MM cells. 450 µL of conditioned media was added to the bottom well in a 24-well plate. H929 cells (1 × 10^5^) were washed three times in serum-free media and seeded in 300 µL in a cell culture insert (Millicell Cell Culture Insert, 12 mm, polycarbonate, 8.0 µm, Merck, Rahway, NJ, USA) in each well. Cells were incubated at 37 °C for 4 h. The cell culture inserts were removed from each well and the media from the lower well was collected and centrifuged at 450× *g* for 3 min. Cell pellets were resuspended in 100 µL dPBS and cell number was determined by flow cytometry. Cells were analysed at a flow rate of 30 µL/min for 90 s through the Cytoflex LX. The cell population was gated using a control sample and the number of migrated cells was determined based on this gate.

### 2.13. Proliferation Assay

THP-1 conditioned media, isolated following 24 h co-incubation with H929, U266 or MM.1S EVs was used to examine the proliferation of H929 cells. H929 cells were seeded at 2.5 × 10^4^ cells/100 µL conditioned media in a 96-well plate that was pre-coated with Poly-D-lysine (Gibco). The cells were incubated at 37 °C for 72 h in an Incucyte S3 and images were captured every 3 h for proliferation analysis using the non-adherence cell analysis software. Cell number was normalised to T_0_ = 3 h and fold change after 72 h was examined.

### 2.14. Preparation of Samples for Mass Spectrometry

Samples were diluted in 8 M Urea/50 mM Tris-HCL with phosphatase and protease inhibitors (Abcam). Samples were sonicated (sonicator 3000, Misonix Inc., New York, NY, USA) for 3 × 8 s and stored on ice. Protein concentration was determined using nanodrop (Nanodrop 2000, ThermoFisher) protein A280. All samples were normalised to 50 µg. Samples were reduced by adding 8 mM dithiothreitol (Fisher Scientific, USA) and incubated at 30 °C shaking (1000 RPM, Thermoshaker Comfort, Eppendorf, Hamburg, Germany). Samples were then carboxylated by adding 20 mM iodoacetamide (VWR, Radnor, PA, USA) and incubated at 30 °C for 30 min shaking (1000 RPM) in the dark. Samples were diluted using 50 mM Tris-HCL to reduce urea concentration to below 1 M. Samples were digested overnight using 1 µg Trypsin (Pierce, Batavia, IL, USA) shaking (850 RPM, 37 °C, Thermoshaker). Samples were diluted to 1% formic acid to terminate digestion.

Sample clean-up was carried out using C18 columns (HyperSep^TM^ SpinTip, Fisher Scientific). C18 columns were activated using elution buffer (60% acetonitrile (Merck, Darmstadt, Germany) in 0.1% Tri-fluoric acid (Merck)) and centrifuged at 664× *g*. The columns were then equilibrated using wash buffer (0.1% Tri-fluoric acid) and centrifuge to pass liquid. The sample was passed through the C18 column, and the column was then washed using a wash buffer. Samples were eluted using elution buffer into 150 µL. The elution was then evaporated off using an Eppendorf concentrator at 30 °C until the peptide pellet was dry.

### 2.15. Mass Spectrometry and Analysis

Peptides were resuspended in 0.1% formic acid and loaded onto an Aurora UHPLC column (25 cm × 75 μm ID, C18, 1.6 μm) (Ionopticks, Fitzroy, Australia). Samples were run on a Bruker timsTof Pro (Bruker, Bremen, Germany) mass spectrometer connected to a Bruker nanoElute nano-lc chromatography system. A 1 h acetonitrile gradient was run at a flow rate of 250 nL/min at 45 °C in positive ion mode. The mass spectrometer was set to all data acquired with the instrument operating in trapped ion mobility spectrometry (TIMS) mode. Trapped ions were selected for ms/ms using parallel accumulation serial fragmentation (PASEF). A scan range of (100–1700 *m*/*z*) was performed at a rate of 10 PASEF MS/MS frames to 1 MS scan with a cycle time of 1.16 s. Capillary voltage of 1500 V, dry gas flow of 3 L/min and a dry temperature of 180 °C.

The raw data was searched against the Homo sapiens subset of the UniProt Swissprot database [22] (reviewed) using the search engine Maxquant [23] (release 1.6.14.0) using specific parameters for trapped ion mobility spectra data-dependent acquisition (TIMS DDA). Each peptide used for protein identification met specific Maxquant parameters, i.e., only peptide scores that corresponded to a false discovery rate (FDR) of 0.01 were accepted from the Maxquant database search. The normalised protein intensity of each identified protein was used for label-free quantitation (LFQ).

Analysis was carried out on LFQ intensity values using Perseus [24] (1.6.13.0). Data was filtered based on only identified sites and removing all possible contaminants and reverse sequences. Data was transformed in log2(FC), and imputation was carried out based on normal distribution. Valid hits were restricted to peptides present a minimum of 50% of at least one group. The fold change cut-off was set at 2 and the P-value cut-off at 0.05. Plots and further analysis were carried out on https://www.bioinformatics.com.cn/en (accessed on 5−14 April 2023) and https://provision.shinyapps.io/provision/ (accessed on 14 April 2023).

### 2.16. Statistical Analysis

Plots are represented by at least 3 independent replicates with the mean and standard error of the mean (SEM) shown. Statistical significance was determined by *p*-value < 0.05. (0.01 < *p* value (∗) < *p* value (∗∗) < 0.01, *p* value (∗∗∗) < 0.001). GraphPad Prism 10 was used to carry out statistical analyses.

## 3. Results

### 3.1. Confirmation of EV Isolation from MM Cell Lines

EVs from H929, U266 and MM.1S cells were isolated from the culture supernatant using ultracentrifugation and density gradient ultracentrifugation. EV confirmation was carried out following the MISEV2018 guidelines [25]. The purified EVs from cell line conditioned media (CM) were visualised using transmission electron microscopy (TEM). Using TEM (Figure 1A), the traditional EV-associated cup morphology and size complement were observed in the purified EV fraction.

The size profile of each cell line derived EV population was determined via nanoparticle tracking analysis (NTA) (Figure 1B). EVs were isolated from 120 mL of CM after 48 h in culture, and cell viability was assessed to ensure cells remained viable during the incubation (Appendix A). NTA results reveal that the mode diameter of the particles varied between cell lines, with an overall mode diameter between 125.2 and 150.3 nm. These results indicate that the EVs isolated from H929, U266 and MM.1S cell lines are enriched for small EVs (sEV) (less than 200 nm diameter). Importantly, even though each cell line was cultured at equal density and volume, results reveal a significant difference in the EV number released per cell line over the 48 h in culture. Specifically, U266 cells (6.50 × 10^10^ EVs) released almost twice as many EVs as H929 cells (3.92 × 10^10^ EVs) and three times more EVs than MM.1S cells (1.83 × 10^10^ EVs). Overall, these results reveal that the MM cell lines examined release EVs at different rates, with U266 cells releasing more vesicles compared to H929 and MM.1S cells, whereas the MM.1S released the lowest quantity of EVs.

Western blot analysis was carried out to confirm EV isolation following the MISEV2018 guidelines [25]. The CD147 membrane protein was found to be present on all cell line derived EVs, whereas the CD9 membrane protein was detected on MM.1S EVs but was undetectable on H929 and U266 derived EVs (Figure 1C). CD147 is known to be glycosylated resulting in a higher molecular weight dispersed banding pattern detected in the H929 EVs, whereas a low-glycosylated form is detected in U266 and MM.1S cell lines and their released EVs. Thus, the variation in CD147 glycosylation is cell-dependent. All cell and EV extracts were positive for the cytosolic or periplasmic protein, TSG101. Interestingly, an additional higher molecular weight form of TSG101 was detected in cell extracts which is likely due to a post-translational modification such as monoubiquitination as previously reported [26]. An additional intracellular marker (HSP70) associated with EVs was identified in U266 and MM.1S EVs. Finally, we detected low levels of the endoplasmic reticulum (ER) protein, HSP90B1, which is generally not associated with sEVs supporting the NTA data that MM cell lines secrete sEVs.

### 3.2. EV Uptake by THP-1 Monocyte Cells Is Dose and Cell Line Dependent

Next, we aimed to mimic the MM TME by co-culture of monocytes and EVs. Specifically, THP-1 monocyte cells were co-incubated for 24 h with a range of EV concentrations (25–400 EVs/cell) derived from either H929, U266 and MM.1S cells, and pre-stained with the fluorescent dye, frCellTrace. The frCellTrace positive cells that are indicative of EV uptake were assessed by flow cytometry and revealed that THP-1 cells take up EVs from all MM cell lines tested in a dose-dependent manner (Figure 2A). Interestingly, fluorescence values indicate that EV uptake by THP-1 cells varied depending on the cell line of origin and this is most apparent at the highest EV dose (400 EVs/cell) where H929 EVs were taken up by 94.56% of cells, whereas U266 and MM.1S EV uptake occurred at a lower frequency of 74.66% and 75%, respectively.

The quantity of EVs taken up by each cell was determined by analysing the median fluorescent intensity (MFI) of the THP-1 cells following co-culture (Figure 2B), as the fluorescence intensity is correlated to the number of EVs present in or on the cell.

Results indicate that the number of EVs taken up by THP-1 cells is also cell line dependent. For example, a comparison of the EV uptake data following incubation with 400 EVs/cell revealed an average MFI of 137,509.67 for H929 EVs, whereas a significantly lower MFI was detected for U266 (32,160.5 MFI) and MM.1S (10,828.83 MFI) EVs. As these results indicate that H929 EVs have a significantly higher rate of interaction with THP-1 cells and/or are taken up more readily by THP-1 cells than U266 or MM.1S EVs. Overall, this data demonstrates that monocyte cells are capable of taking up EVs from MM cells and factors specific to the distinct EV populations may influence uptake efficiency by THP-1 cells. The uptake of H929 EVs by THP-1 cells was confirmed by confocal microscopy and demonstrates that EVs are present in the cell cytosol and on the cell membrane (Figure 2C). Moreover, Z-stack images confirm EV internalisation and dispersion throughout the recipient cell cytosol (Figure 2D). Additionally, U266 and MM.1S EV uptake was also established through confocal microscopy (Appendix A).

### 3.3. H929 EV Uptake by THP-1 Cells Is Energy-Dependent

Finally, to investigate whether EV uptake occurs via an energy-dependent or passive process, uptake assays were performed at 4 °C, where energy-dependent processes are inhibited, but not protein interactions. THP-1 cells were incubated with frCellTrace labelled H929 EVs at 37 °C and 4 °C for 4 h and the MFI of the THP-1 cells was determined by flow cytometry. Results reveal an increase in MFI consistent with EV uptake when incubated at 37 °C, which was inhibited when incubated at 4 °C (Figure 2E), indicating that the process of H929 EV uptake by monocytes is energy-dependent.

### 3.4. Differential IL-6 and MMP-9 Secretion from THP-1 Cells following Co-Culture with MM-Derived EVs

Analysis of IL-6 and MMP-9 secretion by THP-1 cells was carried out following EV uptake to determine the ability of MM-derived EVs to modify the recipient cell behaviour towards a pro-tumourigenic phenotype. THP-1 conditioned media was collected following EV uptake and IL-6, and MMP-9 levels were quantified by ELISA. Results indicate that H929 and U266 EV uptake increased IL-6 and MMP-9 secretion from monocytes in an EV dose-dependent manner (Figure 3A,B), however, significant differences were detected between the EV populations. THP-1 cells incubated with H929 EVs (400 EVs/cell) stimulated a 300-fold increase in IL-6 and a 35-fold increase in MMP-9 whereas U266 EVs displayed a marginal response. Cells co-cultured with MM.1S EVs showed no notable increase in IL-6 secretion and while there was an increase in MMP-9 secretion it was not dose-dependent. Notably, the increased secretion of IL-6 and MMP-9 by monocytes following uptake of H929 EVs compared to U266 and MM.1S EVs is consistent with data from Figure 1 demonstrating that THP-1 cells take up H929 EVs more readily than U266 and MM.1S EVs. Collectively these data suggest that increased IL-6 and MMP-9 secretion by THP-1 cells correlates with EV uptake. Interestingly, when the EV uptake is normalised between 2 distinct populations (Figure 2B H929 EVs (200 EV/cell) and U266 EVs (400 EV/cell), the resulting pro-inflammatory signals are still greater from monocytes following culture with H929 EVs (Appendix A).

### 3.5. MM-Derived EVs Alter THP-1 Conditioned Media to Promote Migration and Proliferation of MM Cells

The results in Figure 2 and Figure 3 demonstrate that H929 EV uptake by monocyte cells significantly increases the secretion of IL-6 and MMP-9 into the conditioned media, mimicking the TME. Therefore, the ability of MM cells to proliferate and migrate in culture in THP-1 conditioned media containing elevated IL-6 and MMP-9 was investigated to determine if the modified secretion is representative of pre-metastatic niche formation that occurs prior to disease spread. Initially, the proliferation rate of H929, U266 and MM.1S cells was determined in optimal media, RPMI 1640 (10% FBS, 1% pen-strep), over 72 h (Appendix A). Next, cells were incubated in conditioned media isolated from THP-1 cells that were left untreated for 24 h, treated with PBS or EVs (400/cell) for 24 h and cell proliferation was examined after 72 h using Incucyte technology. Cell numbers were normalised after 3 h to ensure that cells adequately settled for camera focusing. Results reveal that H929 cell proliferation is significantly increased when incubated with conditioned media collected from monocytes following co-culture with H929 EVs (Figure 4A). In contrast, no difference in U266 and MM.1S cell proliferation was detected when incubated with conditioned media from monocytes following co-culture with their secreted EVs (Appendix A). This data demonstrates that H929 EVs induce alterations in THP-1 secretome that promotes cell proliferation, whereas U266 and MM.1S EVs do not, and is consistent with greater EV uptake by THP-1 cells (Figure 2A,B) and greater secretion of IL-6 and MMP-9 (Figure 3) demonstrated earlier. To further investigate pro-metastatic alterations induced by H929 EV in the TME, a migration assay was carried out to determine if H929 cells preferentially migrate to the THP-1 conditioned media following uptake of H929 EVs. Results indicate that the migration of H929 towards conditioned media derived from THP-1 cells following EV uptake is significantly increased compared to conditioned media from cells that did not take up EVs (Figure 4B).

Collectively, these results reveal that the altered monocyte secretion generated following H929 EV uptake promotes migration and proliferation of H929 cells and demonstrates that MM EVs promote tumour niche formation by altering monocyte function.

### 3.6. Mass Spectrometry Analysis of MM EV Content

This study so far reveals that MM EVs have a differential effect on monocyte uptake and function, with H929 EVs exhibiting the greatest uptake efficiency and pro-inflammatory modulation within the TME that promotes tumour migration and proliferation. Furthermore, when EV uptake was normalised between H929 EVs (200 EV/cell) and U266 EVs (400 EV/cell), the resulting pro-inflammatory signals were greater following uptake of H929 EVs suggesting that additional factors such as EV cargo may regulate monocyte function and contribute to differential pro-inflammatory signals detected. To investigate this further, mass spectrometry analysis of the three MM cell line derived EV cargo was performed.

Mass-spectrometry based proteomics characterisation of the EVs produced by the different cell lines revealed that all EVs were enriched in protein cargo and contained a similar number of proteins, ranging from 1838 to 2168 proteins (Figure 5A). In order to validate our proteomics approach, we performed several analyses. First, the proteins identified were compared to the Vesiclepedia database of the Top 100 EV proteins, 82 of the Top 100 were identified in all three EV groups and 84 were identified in at least one group, which provides additional confirmation that the samples isolated are EVs. Next, 2D Principal Component Analysis (PCA) was applied to the dataset to identify replicate similarity as well as show sample grouping. Encouragingly, replicate EVs from each cell of origin grouped in clusters, indicating an adequate standard of biological and technical replicates (Figure 5B). Among the different EVs of origin, group differences are clearly observed when separated between the two principal components. This is further demonstrated in the heatmap where samples are grouped based on complete clustering using Euclidean distancing (Figure 5C). This analysis suggests U266 EVs have the greatest level of similarity to MM.1S EVs, with H929 EVs differing substantially, which is further displayed in the correlogram using hierarchical clustering of Pearson’s correlation (Figure 6A). All these observations provide confidence that our proteomics approach can be used to characterise differences among the EVs secreted by the different cell lines and we proceeded to perform functional analysis of our dataset.

Differential expression analysis on the three EV proteomic data sets carried out using Perseus, was performed to determine differentially expressed proteins that may contribute to the variation in THP-1 response following uptake of the different MM cell line EVs. Volcano plots (Figure 6B–D) indicate that EV protein levels vary greatly depending on the cell line of origin, with H929 EVs having the largest number of differentially regulated proteins compared to U266 and MM.1S EVs.

During the initial characterisation of EVs (Figure 1) it was determined that CD9 was present in MM.1S EVs at a higher level than H929 and U266 EVs, and this result was reflected in the mass spectrometry data, which also provides a form of retrospective validation. To further validate our findings, we extended the western blot analysis to a panel of selected proteins that are differentially changed. Mass spectrometry data revealed a number of proteins including NRAS, ZAP70, TFAM and CD9 that differ in expression levels across EV groups. Within the mass spectrometry analysis, the levels of GAPDH detected in EVs were not significantly changed among all cell line EVs. Based on this, western blot analysis of selected proteins was carried out and densitometry was performed and normalised to GAPDH. Western blot densitometry data for NRAS, ZAP70 and TFAM are consistent with the mass spectrometry data with significant or near-significant trends detected (Figure 7).

Next, we investigated if we could identify pathophysiological mechanisms associated with EVs from the cell lines. Our results so far demonstrate that MM.1S EVs had no significant effect on monocyte secretion of IL-6 and MMP-9, whereas H929 EVs stimulated a 300-fold increase in IL-6 and a 35-fold increase in MMP-9 and U266 EVs displayed a marginal response (Figure 3). Differential expression analysis was carried out between H929 and U266 EVs as well as U266 and MM.1S EVs, the common significantly upregulated proteins identified through this analysis indicate protein expression that correlates with the response detected in THP-1 cells (i.e., expression levels are highest in H929 EVs and lowest in MM.1S EVs). Results reveal that 15 proteins are highly upregulated in H929 EVs and marginally upregulated in U266 EVs relative to MM.1S EVs, correlating with the response observed in THP-1 cells (Table 1). This positive correlation with increased protein expression, EV uptake and THP-1 secretion response suggests they may have some involvement in THP-1 immune response to EVs. Of note, of the 15 proteins, a number are involved in increased exocytic secretion (RAB8b, Exoc4), splicing (FUS, SRPK2) and focal adhesion (integrins ITGB1, ITGA4), which is associated with cancer progression.

In light of our previous observations, further analysis of the differential expression of MM.1S and U266 EV proteins was carried out against H929 EVs to identify proteins or pathways that are majorly upregulated resulting in the largest response. Subsequently, KEGG pathway analysis was carried out to determine the downstream effects of the differentially regulated proteins. KEGG analysis shows a similar output of enrichment for H929 EVs over U266 and MM.1S EVs (Figure 8A,B). The results indicate the enrichment is predominantly focused on transcriptional regulation with RNA splicing and the spliceosome being heavily enriched (Figure 8D). Additionally, there is an enrichment for the T-cell receptor signalling pathway, VEGF signalling and a number of other cancer-related pathways. Downregulated pathways include metabolisms such as glycolysis/gluconeogenesis, amino sugar and nucleotide sugar metabolism and carbon metabolism. Of interest also is the finding that apoptosis and necroptosis pathways were downregulated in H929 EVs compared to the other two cell line EVs (Appendix A). Furthermore, the KEGG pathways enrichment for U266 EVs over MM.1S EVs (Figure 8C) shows enrichment for endocytosis, phagosome and the RAS signalling pathways, while the major enrichment for the spliceosome detected in H929 EVs is not present.

## 4. Discussion

A large number of studies suggest that cancer cell EVs contribute to the formation of a tumour niche, aid in metastasis, suppress the active and innate immune system, generate therapeutic resistance and contribute to overall cancer progression [18,19,27,28,29,30,31]. To explore the effects of MM-derived EVs on monocyte cells, a comparison of THP-1 cells uptake of 3 different MM cell line derived EV populations revealed that while in all cases EVs were taken up by the monocyte cells in a dose-dependent manner, the EV cell line of origin greatly influenced the level of uptake. While U266 and MM.1S derived EVs had a similar uptake by the THP-1 cells, H929-derived EVs were taken up by more cells and at a significantly greater number per cell. This data suggests that EV factors play a role in controlling EV uptake. Confocal microscopy of cells following co-culture with fluorescently labelled EVs confirmed internalisation within the monocyte cell cytoplasm.

EV uptake has been previously demonstrated to occur as energy-dependent endocytosis following co-incubation of cells and EVs at 37 °C, which can be inhibited at 4 °C [32,33,34]. Consistent with that, results from this study reveal H929 EV uptake by monocytes at 37 °C, which can be abolished at 4 °C. Endocytosis is the process by which cells actively internalise external components to the cell, including drugs, molecules and EVs and includes non-selective processes of phagocytosis and pinocytosis, and the selective process of receptor-mediated endocytosis [35]. Confirmation of energy-dependent endocytosis as the primary route of EV uptake suggests that EV membrane proteins are responsible for EV-cell interaction, which mediates ATP-dependent receptor-mediated endocytosis in THP-1 cells [36,37].

While MM cells have been shown to produce IL-6, the major source occurs through the BM microenvironment [11]. Within the BMM, monocytes release IL-6 and stimulate signal secretion by other cells such as LPS, IL-1, TNF, and platelet-derived growth factor [12,38]. Previously it has been shown that the co-culture of BM microenvironment cells with MM cells results in an upregulation and increased release of IL-6 by the BM microenvironment cells [39,40,41,42]. Previous studies have reported on the role of IL-6 in MM progression using osteoclast-, macrophages- and stromal cells-MM cells co-cultures; However, immature monocyte-specific effects have not been previously investigated. Additionally, these studies focus on MM cells, thus the role MM-derived EVs have on the regulation of IL-6 secretion remains unknown. While the role of MMP-9 in MM is clear, the role of EVs in the release of MMP-9 into the microenvironment remains poorly understood. Results from this study show that THP-1 secretion of IL-6 and MMP-9 can be modulated by MM-EVs. The rates of secretion of IL-6 and MMP-9, particularly mediated by H929 and U266 derived EVs, occurred in a dose-dependent manner and were increased most significantly when co-incubated with H929-derived EVs, whereas MM.1s-derived EVs have no significant effect. These results indicate that the EV cell of origin plays an important role in EV uptake by THP-1 cells, as well as in the downstream functions of the recipient cell such as secretion of IL-6 and MMP-9. The release of these two key factors produces an environment receptive to myeloma cell survival and progression. Of the MM-derived EVs examined, only H929 EVs significantly increased the levels of IL-6 in the THP-1 conditioned media, which reached a 300-fold increase. While these finding simprove our understanding of EV signalling within the TME, the work was carried out on a monocyte cell line and future studies using primary monocytes would provide validation of the effects.

The ability of cancer cells to migrate and proliferate at a new location is a hallmark of disease progression and metastasis. Our results show that MM cells preferentially migrate towards, and undergo increased proliferation, in conditioned media produced by THP-1 cells following EV uptake, which contains higher levels of Il-6 and MMP-9, and presumably other unidentified factors, confirming that EVs modulate the monocyte secretome to produce a pro-metastatic environment. Collectively, this data suggests that EVs play a role in pre-metastatic niche formation and MM cell dissemination, migration, and proliferation at a new site.

A major contribution of our study is the identification of the interactome of the EVs of these cell lines using mass-spectrometry proteomics. Bioinformatic analysis of mass spectrometry data revealed differentially enriched EV protein cargo across the three EV populations. Our datasets are of high quality as confirmed by subsequent experimental validation of the mass spectrometry data carried out using western blot analysis of a selection of targets including ZAP70, NRAS, TFAM, and CD9. Western blot data for the target proteins correlated with the mass spectrometry results thereby validating the EV proteomic profiles produced.

Analysis of the proteomic data may provide an explanation for differential effects on the immune cells. For example, when considering the rate of uptake and THP-1 secretion between the different MM-derived EVs, an analysis was carried out to determine proteins that were upregulated in a similar pattern to the levels of uptake and secretion (i.e., lowest in MM.1S and highest in H929). This analysis revealed a number of proteins including ITGB1 and ITGA4, that make up the integrin α4β1 (VLA-4) complex, which is increased in expression in MM patients and correlates with disease progression [43]. VLA-4 on myeloma cells binds with high affinity to its ligands and promotes MM cell adhesion and survival within the BM microenvironment [44]. Additionally, the interaction of VLA-4 and VCAM-1 on bone marrow microenvironment cells results in the reduction of osteoblastogenesis and increases osteoclastogenesis which drives bone disease [45]. Furthermore, monocytes in MM tumour-bearing mice have a significantly higher level of VLA-4 expression than their non-tumour-bearing counterparts [43]. VLA-4 expression on monocytes not only supports the migration of inflammatory monocytes but also activation of Rac2 on macrophages leading to the transition of classically activated (M1) macrophages to immunosuppressive M2 cells [46]. As EVs are representative of their cells of origin, it is expected that the highly expressing VLA-4 cells will produce EVs enriched with VLA-4. High expression levels of VLA-4 in H929 EVs, which positively correlates with altered secretome of monocytes may contribute to pre-metastatic niche formation in MM.

Importantly, our results may also shed light on the mechanistic and functional aspects of MM cells. This is exemplified by KEGG pathway analysis indicating a significant increase in pathways such as the spliceosome, T-cell receptor signalling, DNA replication and mRNA surveillance pathways in H929 EVs, compared to U266 and MM.1S EVs. The spliceosome is a large protein-rich complex responsible for removing introns from the pre-mRNA generated in the cell to produce mature messenger RNA (mRNA). The spliceosome process is essential for cell function and dysregulation or mutation of components of the spliceosome complex has been identified as a driver of many forms of cancers [47]. Modification in the expression levels of components of the spliceosome complex results in alternative splicing. Alternative splicing events result in changes in protein expression that can lead to cancer-associated phenotypes through the promotion of angiogenesis, cell proliferation and anti-apoptotic behaviour [48]. For example, alternative splicing of MCL1 exon 2 results in the pro-apoptotic MCL1S or anti-apoptotic MCL1L [49]. As EVs are representative of the cell of origin, the finding that H929 EVs are heavily enriched for spliceosome-related proteins suggests that the H929 cell proteome may undergo more alternative splicing events compared to the other cell lines. Moreover, the transfer of multiple splicing factors to monocytes via EVs revealed in this study raises the possibility that deregulated splicing contributes to the altered monocyte function towards a pro-metastatic phenotype. In support of that, studies have shown that apoptotic cell derived EVs promote malignancy of glioblastoma through the intracellular transfer of splicing factors thereby promoting therapy resistance and an aggressive migratory phenotype in the recipient cells [50]. This indicates that the increased levels of spliceosome components packed into EVs drive the progression of disease. Furthermore, inhibition of key spliceosome factors (SF3B1, SF3A1, SF3A2, SF3A3, U2AF1, SRSF2, or EFTUD2) in mouse and human macrophages blocks the toll-like receptor (TLR)-induced cytokine production through limiting NF-ĸB activation. The role of these factors in pro-inflammatory signalling is further clarified by the study showing that expression of SF3B1, U2AF1, and SRSF2 containing common mutations involved in myeloid dysplastic syndrome results in an enhanced level of NF-ĸB activation and subsequently increased cytokine response [51,52,53,54,55]. Interestingly, the absence of miR-16 in MM-EVs was shown to enhance NF-ĸB activation in mature primary monocytes resulting in the polarization of monocytes to M2 macrophages [56] Collectively, these studies confirm that MM EV components regulate cytokine signalling in recipient cells and is in agreement with this study.

In conclusion, this study reveals that multiple myeloma cell line EVs undergo differential uptake by monocytes, which is dependent on the cell line of origin. Moreover, EV uptake by monocytes results in an altered secretome that enhances the migration and proliferation of myeloma cells supporting the role of EVs in tumour niche formation. Finally, EV proteome analysis reveals potential mechanisms by which EV cargo mediates pro-metastatic alterations within the tumour microenvironment.

## Figures and Tables

**Figure 1 cancers-16-01011-f001:**
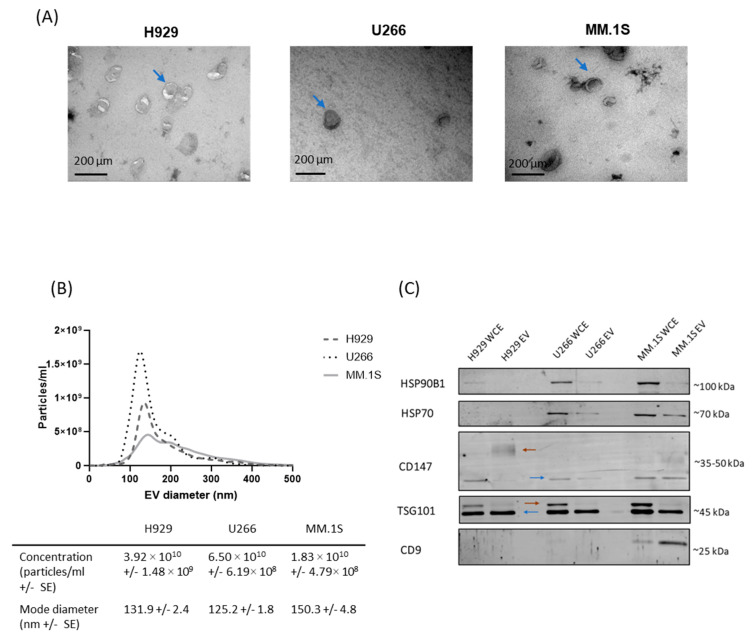
Confirmation of MM-derived EV isolation. (**A**) Representative negative staining transmission electron microscopy of MM cell line-derived EVs. Scale bar = 200 µm. Blue arrows are included to indicate EVs. (**B**) Nanoparticle tracking analysis of the MM cell line derived-EVs (diluted 1:100–1:200) was carried out to determine their concentration and size distribution. The table indicates concentration (particles/mL) and particle mode diameter (nm) of the final EV samples. (**C**) 3 µg of protein/cell line and EV sample was resolved on a 12% polyacrylamide gel and probed with primary antibodies (HSP90B1, HSP70, CD147, TSG101, CD9). The blue arrow indicates native protein while orange arrow indicates post-translational modification.

**Figure 2 cancers-16-01011-f002:**
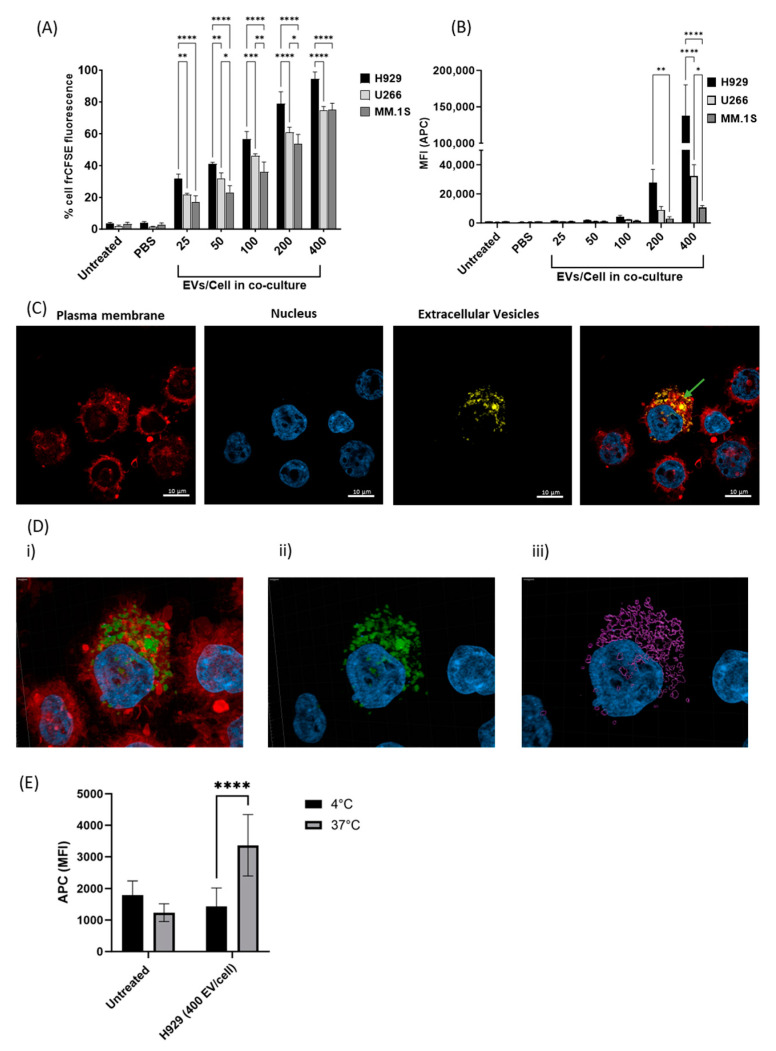
Dose dependent yet differential uptake of MM-derived EVs by THP-1 monocytes is dependent on cell line of origin and is an energy-dependent process. MM-derived (H929, U266, MM.1S) EVs were stained with the membrane permeable dye frCellTrace and co-incubated with THP-1 cells (2.5 × 10^5^ cells/mL) at a range of concentrations (25–400 EVs/cell) for 24 h. (**A**) The percentage of cells showing inherited frCellTrace staining following co-incubation with MM-derived EVs. (**B**) The intensity of EV uptake was determined by the median fluorescent intensity of the frCellTrace positive THP-1 cells following treatment. (**C**) THP1 cells were stained with Cell Mask Orange Plasma membrane stain and DAPI and confocal microscopy was carried out using Zeiss LSM 800 Airy. Green arrow; indicates the location of MM EVs. (**D**) 3-dimensional depiction of H929-EVs location within the THP1 cell is shown using Z-stack imaging. Nucleus (Blue) included to aid in highlighting EV internalisation. (**i**) plasma membrane (cell mask orange) staining and EV (frCellTrace) staining. EVs alone in their (**ii**) fluorescence and (**iii**) 3D composition. (**E**) THP-1 cells were co-incubated with H929 EVs at a concentration of 400 EVs/cell and incubated at either 4 °C or 37 °C for 4 h. *p* values: * ≤ 0.05, ** ≤ 0.01, *** ≤ 0.001, **** ≤ 0.0001.

**Figure 3 cancers-16-01011-f003:**
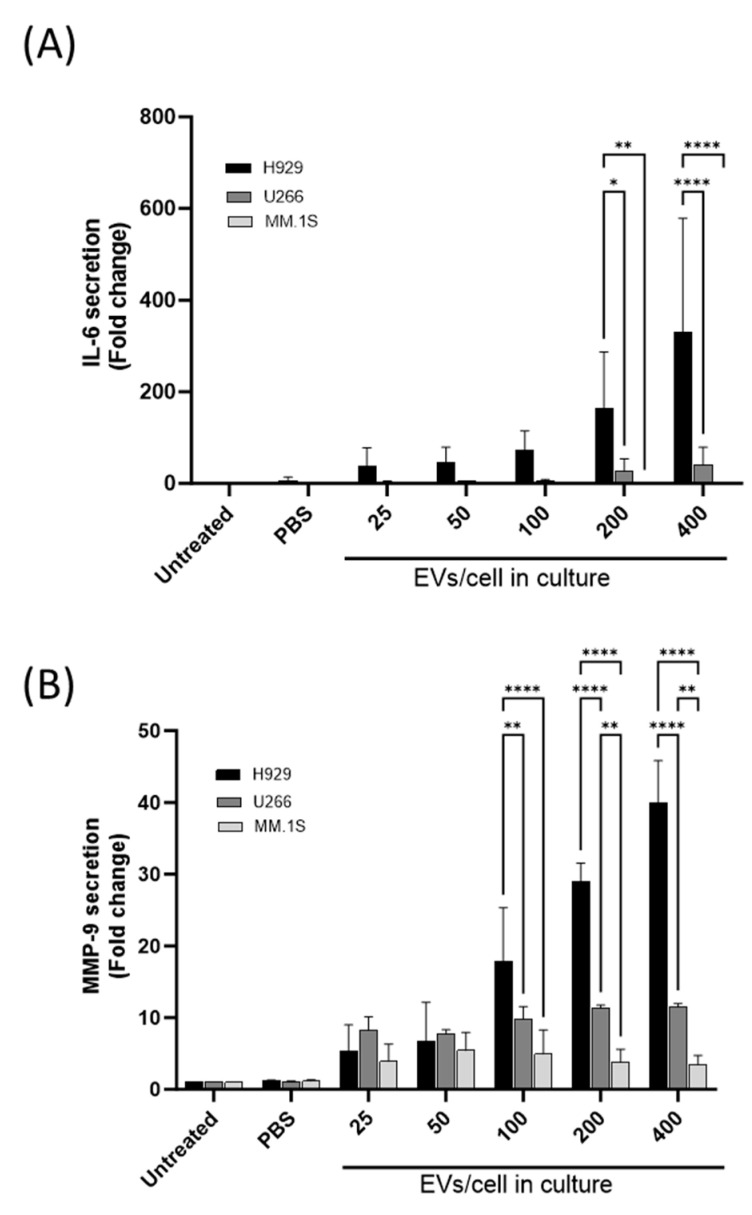
Analysis of IL6 and MMP9 release from THP-1 cells following uptake of MM-EVs. THP-1 cells (2.5 × 10^5^ cells/mL) were co-incubated with various concentrations of MM derived EVs (25, 50, 100, 200, 400 EVs/cell) for 24 h at 37 °C. (**A**) IL-6 and (**B**) MMP-9 levels were determined by ELISA and indicated as fold change over untreated samples. *n* = 3, *p* values: * ≤ 0.05, ** ≤ 0.01, **** ≤ 0.0001 two-ANOVA with Turkey’s multiple comparisons. Error bars indicated SME for independent experiments.

**Figure 4 cancers-16-01011-f004:**
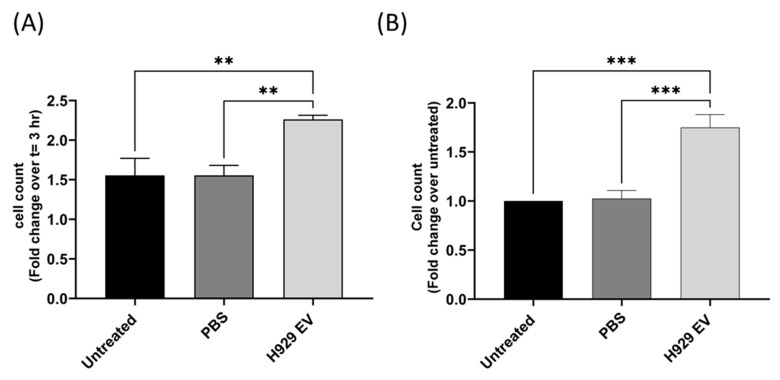
MM H929 cell proliferation and migration in THP-1 conditioned media following uptake of MM-H929 EVs. Conditioned media (CM) was collected from THP-1 cells that were left untreated or co-incubated with PBS or H929 EVs (400 EVs/cell) for 24 h. (**A**) 2.5 × 10^4^ H929 cells were incubated in THP-1 CM collected following treatment (untreated, PBS, H929 EV) for 72 h. Cell proliferation was measured at 72 h using Incucyte^TM^ technology and expressed as fold change over the 3 h time point. (**B**) H929 cells (1 × 10^5^) were incubated in 300 µL RPMI 1640 (1% pen/strep) in the upper section of a Boyden chamber with 450 µL THP-1 conditioned media in the lower chamber for 4-h. The number of migrated cells were counted using flow cytometry. Error bars indicate SEM of three independent experiments. *n* = 3, One-way ordinary ANOVA *p*-value; ** ≤ 0.01, *** ≤ 0.001.

**Figure 5 cancers-16-01011-f005:**
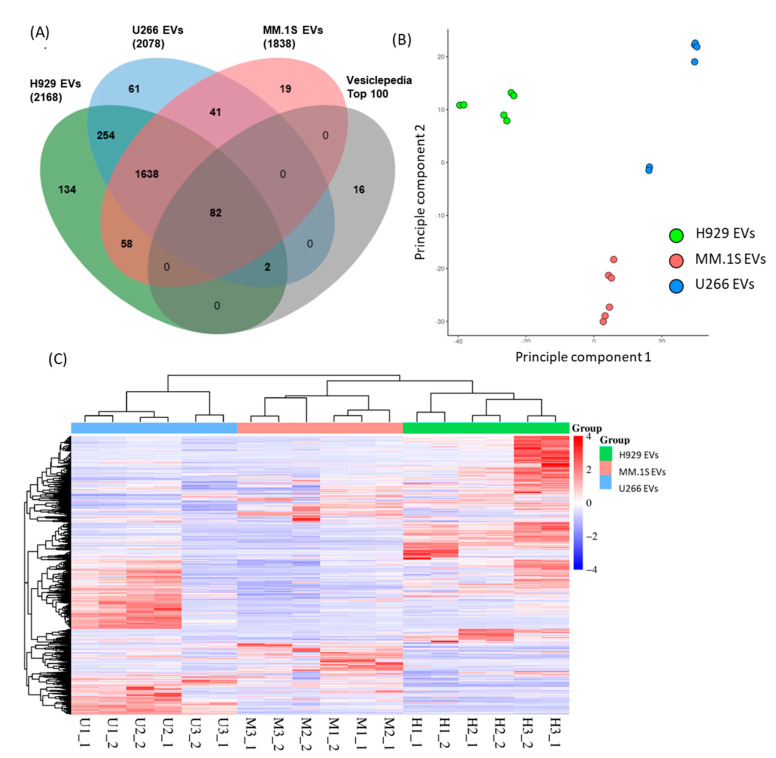
Comparative analysis of H929, U266 and MM.1S derived EVs proteome. (**A**) Venn diagram comparing all protein hits identified in H929, U266 and MM.1S EVs and the top 100 proteins identified on Vesicleopedia. (**B**) PCA plot using the two highest principal components to group each MM derived EVs. (**C**) Heatmap showing complete proteome of MM-derived EVs. Sample distances were determined by the correlation of samples. PCA was plotted by https://provision.shinyapps.io/provision/ (accessed on 14 April 2023) and the Venn diagram and heatmap were plotted by https://www.bioinformatics.com.cn/en (accessed on 5–14 April 2023), free online platforms for data analysis and visualization.

**Figure 6 cancers-16-01011-f006:**
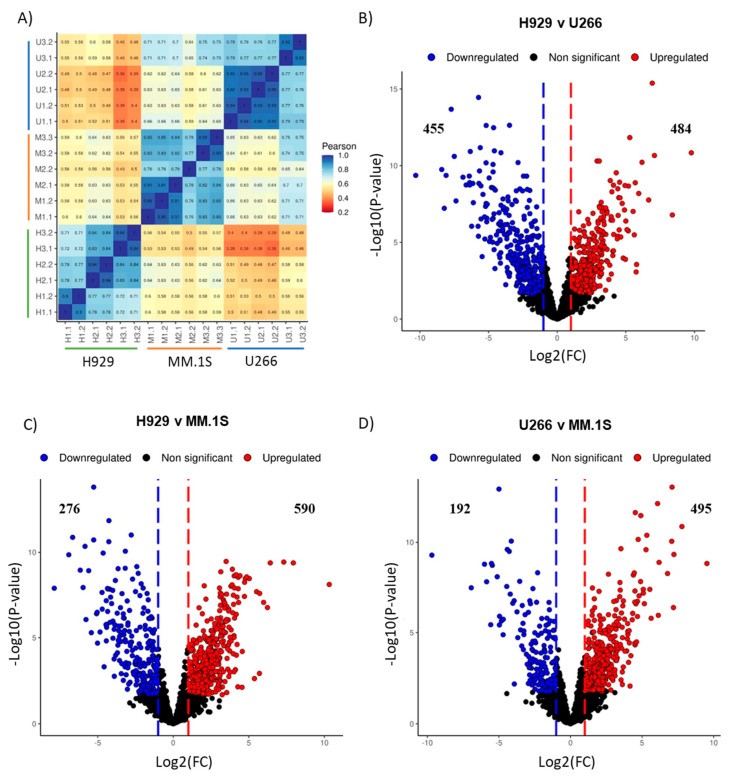
A deep analysis of MM-derived EVs proteomics. (**A**) Hierarchical clustering of Pearson correlations. Correlation within groups was <0.7, while correlation varied greatly between groupings dropping to as low as 0.36. (**B**–**D**) Volcano plots of quantified protein changes from MM-derived EVs. The line indicates a log 2-fold change (FC) of protein abundance, indicated on the horizontal axis. The *p*-value is indicated by the −log 10 of the *t*-test, present in the vertical axis. Significantly up-regulated proteins are shown in red and down-regulated proteins are shown in blue. Insignificant proteins are indicated in black. Statistical cut-off was set with a *p*-value < 0.05 using Benjamini-Hochberg FDR, Log2(FC) ≥ 1. Figure was developed using https://provision.shinyapps.io/provision/ (accessed on 14 April 2023).

**Figure 7 cancers-16-01011-f007:**
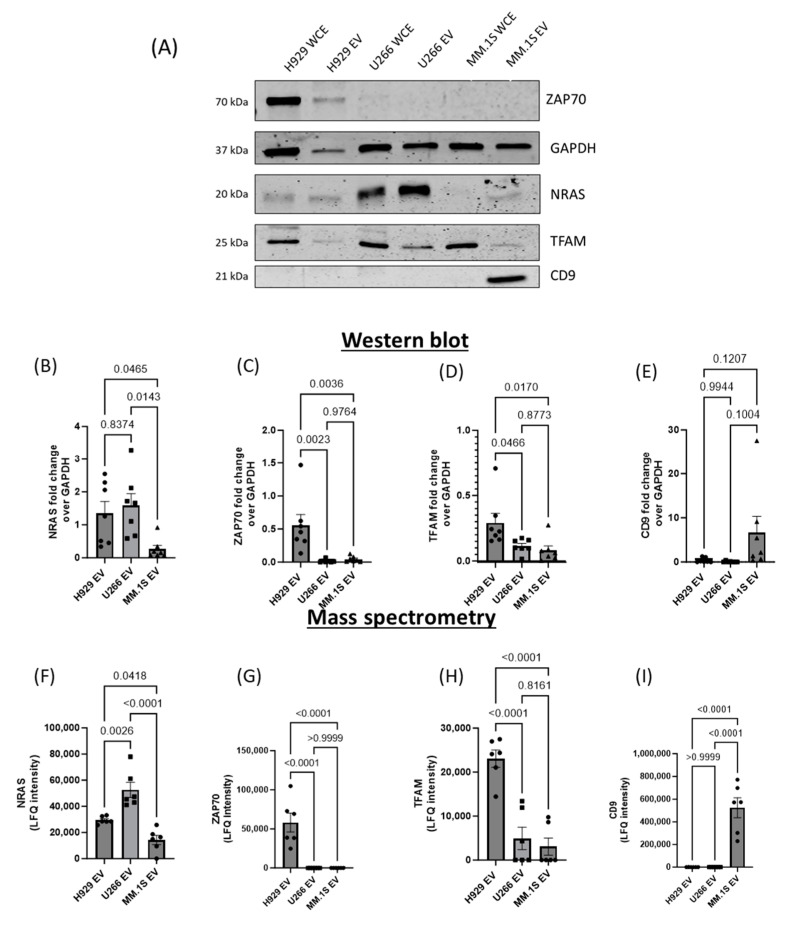
Validation of Mass spectrometry analysis of MM-derived EVs. (**A**) A number of proteins identified as differentially expressed between the MM-derived EVs according to mass spectrometry results were probed for using western blot analysis. 3 µg of both EVs and whole cell extracts (WCE) from the different MM cell lines (H929, U266, MM.1S) were tested for their expression of (**B**) NRAS, (**C**) ZAP70, (**D**) TFAM and (**E**) CD9. GAPDH expression levels were not identified as significant and so all other protein expression levels were normalised to GAPDH across each blot (n = 6). Average LFQ intensities identified through mass spectrometry were plotted for (**F**) NRAS, (**G**) ZAP70, (**H**) TFAM and (**I**) CD9 for comparison. Statistical analysis: One-way ordinary ANOVA. Error bars = standard median error.

**Figure 8 cancers-16-01011-f008:**
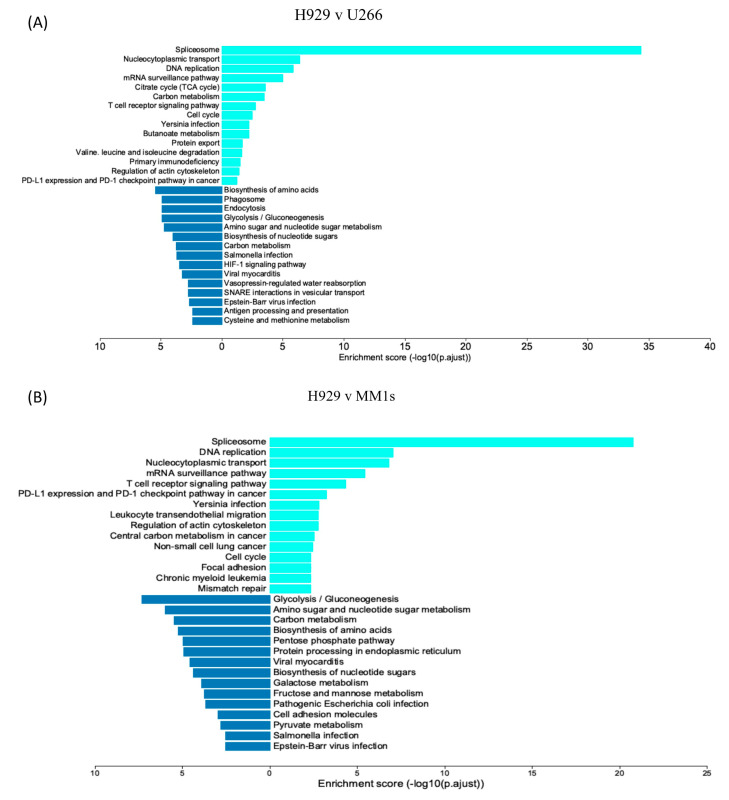
KEGG pathway enrichment analysis of H929, U266, MM.1S EV proteomics. A comparative analysis of MM-derived EV proteomics was carried out. The top 15 pathways both positively and negatively enriched based on the differential gene expression of (**A**) H929/U266 EVs, (**B**) H929/MM.1S EVs, (**C**) U266/MM.1S EVs. (**D**) Pathway enrichment of the spliceosome based on H929/U266 EV comparisons shows a high level of protein upregulation (red) and little downregulation (green).

**Table 1 cancers-16-01011-t001:** Proteins with expression levels in H929, U266 and MM.1S EVs that positively correlate with THP-1 EV uptake level, and IL-6 and MMP-9 secretion.

	H929/U266		U266/MM1s		H929/MM1s	
	log2 (FC)	FDR	log2 (FC)	FDR	log2 (FC)	FDR
MYL4	3.89384	8.03 × 10^−7^	1.31732	0.002187	5.211161	6.02× 10^−10^
H1F0	3.390922	3.47× 10^−5^	1.402214	0.022585	4.793137	9.78× 10^−7^
VAT1	2.964245	0.000402	4.253141	6.16× 10^−05^	7.217386	8.10× 10^−7^
ITGB1	2.866007	0.000122	2.102115	0.002692	4.968122	1.99× 10^−6^
FUS	2.791342	0.000183	1.567522	0.004318	4.358864	8.27× 10^−7^
TF	2.231458	0.043132	2.657278	0.01826	4.888736	1.73× 10^−7^
ITGA4	2.120799	0.000634	4.002313	0.002133	6.123113	5.79× 10^−5^
MANF	1.975773	0.01541	2.34326	0.005678	4.319033	1.79× 10^−6^
SRPK2;SRPK3	1.822525	0.040219	2.117563	0.001355	3.940088	0.000108
KIF4A	1.800351	5.92× 10^−05^	1.826726	0.005663	3.627077	2.95× 10^−5^
MAPRE2	1.676595	0.004901	1.742989	0.006958	3.419584	1.98× 10^−5^
NDRG3	1.543764	0.011512	1.886626	0.050681	3.43039	0.000854
HIST1H1D	1.417215	0.000121	2.381263	0.029233	3.798478	0.002065
EXOC4	1.184284	0.01465	1.865372	0.010685	3.049656	7.39× 10^−5^
RAB8B	1.104097	0.005696	1.514335	0.02753	2.618432	0.00118

## Data Availability

The mass spectrometry proteomics data will be deposited to the ProteomeXchange Consortium via the PRIDE [57] partner repository with the dataset identifierPXD047285.

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
