# Peer review of "Multiple Myeloma Derived Extracellular Vesicle Uptake by Monocyte Cells Stimulates IL-6 and MMP-9 Secretion and Promotes Cancer Cell Migration and Proliferation"

_cancers, 2024, doi:10.3390/cancers16051011_

Round 1

Reviewer 1 Report

Comments and Suggestions for Authors

In this manuscript, the authors have identified that extracellular vesicles from human myeloma cell lines can increase the secretion of IL-6 and MMP-9 from monocytes. The authors have also used proteome analysis to identify differential cargo enrichment in myeloma EVs that may contribute to myeloma disease progression. The methods and data are well explained and detail-rich. The authors can improve the significance study by providing certain details as outlined below:

Major points:

1)    What is the mechanism behind EVs increasing IL-6/MMP-9 secretion from THP-1 cells? Comparing differential cargo between H929 and MM1.s, there are several interesting potential pathways to explore. Loss-of-function experiments can further pin down this mechanism.

2)    Figure 4A: Although the cell count increase is significant, it is a moderate increase. Longer time points or increased dosage of EVs could be explored. A line graph for cell proliferation over time would be useful for understanding kinetics.

3)    Figure 6 B, C, D: top genes/genes of interest can be indicated on the volcano plot 

4)    Clarification: Fig 7A: does the NRAS antibody recognize both mutant and wild-type protein?

5)    Discussion: The authors must address the limitations of this study, e.g., use of THP-1 cells vs CD14+ monocytes etc.

6)    Discussion: There are several other studies that have looked at myeloma derived EVs and their effect on macrophages/monocytes, for example: https://doi.org/10.1182/bloodadvances.2021006187,https://doi.org/10.1172%2Fjci.insight.129348

Minor points:

1)    Keep cell line, protein names consistent (include hyphens/period) e.g., THP-1, IL-6, MM1.S etc.

2)    Line 448: Is that a typo of two-way ANOVA?

3)    Fig 2C legend must include what the green arrow points to

4)    Fig 5B: (Typo) Principal component

5)    Fig 8A, B, C: X axis: typo in p.adjust

6) Fig 8D: what parameter does the scale bar represent?

Comments on the Quality of English Language

Overall the manuscript is well written and easy to understand. However, a few typos need to be corrected:

1)    Keep cell line, protein names consistent (include hyphens/period) e.g., THP-1, IL-6, MM1.S etc.

2)    Line 448: Is that a typo of two-way ANOVA?

3)    Fig 5B: (Typo) Principal component

4)    Fig 8A, B, C: X axis: typo in p.adjust

Author Response

In this manuscript, the authors have identified that extracellular vesicles from human myeloma cell lines can increase the secretion of IL-6 and MMP-9 from monocytes. The authors have also used proteome analysis to identify differential cargo enrichment in myeloma EVs that may contribute to myeloma disease progression. The methods and data are well explained and detail-rich.

The authors can improve the significance study by providing certain details as outlined below:

Major points:

1)    What is the mechanism behind EVs increasing IL-6/MMP-9 secretion from THP-1 cells? Comparing differential cargo between H929 and MM1.s, there are several interesting potential pathways to explore. Loss-of-function experiments can further pin down this mechanism.

Response:

While investigating the precise mechanism of action is very interesting, as outlined by the reviewer it loss of function studies will be required to investigate this, which is likely to be extensive and does not guarantee success, and is beyond the scope of the current study.

The authors have speculated/commented on interesting differentially enriched targets that may alter immune cell function.  For example, in the result section it is outlined on line 556 that “of the 15 proteins, a number are involved in increased exocytic secretion (RAB8b, Exoc4), splicing (FUS, SRPK2) and focal adhesion (integrins ITGB1, ITGA4), which is associated with cancer progression”. Furthermore, a paragraph in discussion (line 654) “ ITGB1 and  ITGA4 that make up the integrin α4β1 (VLA-4) complex, is increased in expression in MM patients and correlates with disease progression”. As outlined on line 665 “High expression levels of VLA-4 in H929 EVs, which positively correlates with altered secretome of monocytes may contribute to pre-metastatic niche formation in MM”.

In addition to individual targets above, we have performed KEGG pathway analysis and discussed a potential role of dysregulated spliceosome function. (Paragraph from Line 668). In line 683 it is outlined that “Moreover, the transfer of multiple splicing factors to monocytes via EVs revealed in this study raises the possibility that deregulated splicing contributes to the altered monocyte function towards a pro-metastatic phenotype” and we include supporting evidence that the EV enriched splicing factors are associated with altered cytokine secretion in cancer (refs 51  - 55).

Thus, we have highlighted several interesting potential pathways that could be explored in future studies, however, we remain cautious as to not over speculate as further studies are required to investigate the mechanism of action.

2)    Figure 4A: Although the cell count increase is significant, it is a moderate increase. Longer time points or increased dosage of EVs could be explored. A line graph for cell proliferation over time would be useful for understanding kinetics.

The data in Fig 4 A represents a 1.5 fold increase in proliferation and as outlined by the reviewer, it is significant. The EV dose of 400 EVs/cell was chosen based on data in Fig 2A and B, and Fig 3 A and B, where EV uptake was significant and elicited a significant response in the monocyte cells. 

Cell proliferation data was collected for all cell lines over the course of 72 hours and a line graph displaying proliferation rates at 24, 48 and 72 hr is included in supplementary file (Fig. 5) which may have been missed by the reviewer and is referred to on page 13 under section MM-derived EVs alter THP-1 conditioned media to promote migration and proliferation of MM cells. Data for H929 outlines that proliferation rates are not significantly altered at 24 and 48 hr, however, that is not unexpected given that H929 cells have a doubling time of 50 hours  (https://www.sciencedirect.com/science/article/pii/S0006497120699153)

In agreement with the reviewer, including longer time point or increased dosage of EVs would be interesting, however, it will not change the conclusion of the experiment, which is a significant increase in proliferation, and confirms our hypothesis that EVs alter the tumour microenvironment to promote a metastatic niche.

3)    Figure 6 B, C, D: top genes/genes of interest can be indicated on the volcano plot

The approach taken to compile targets in Table 1 is outlined in Line 549 “common significantly upregulated proteins identified through this analysis indicate protein expression that correlates with the response detected in THP-1 cells (i.e., expression levels are highest in H929 EVs and lowest in MM.1s EVs). Results reveal that 15 proteins are highly upregulated in H929 EVs and marginally upregulated in U266 EVs relative to MM.1s EVs, correlating with the response observed in THP-1 cells (Table. 1).” Therefore, the targets in Table 1 are not the most enriched in single analysis experiments. The identity of top targets in the single analysis is already present in the supplementary data. The authors believe that highlighting the top genes in the single analysis volcano plots will not add to the study.

4)    Clarification: Fig 7A: does the NRAS antibody recognize both mutant and wild-type protein?

Response: The NRAS antibody targets the C-terminal domain and the most common NRAS mutations occur within the N-terminal domain indicating that the antibody can recognise both the mutant and wild-type protein. Antibody details are included under Materials (line 107) (https://datasheets.scbt.com/sc-519.pdf)

5)    Discussion: The authors must address the limitations of this study, e.g., use of THP-1 cells vs CD14+ monocytes etc.

Response: In the revised manuscript the study limitation is included in the discussion – line 629 “While this data is encouraging, the work was carried out on a monocyte cell line and fu-ture studies using primary monocytes would provide validation of the effect that MM EVs have on immature monocyte cells.”

6)    Discussion: There are several other studies that have looked at myeloma derived EVs and their effect on macrophages/monocytes, for example: https://doi.org/10.1182/bloodadvances.2021006187,https://doi.org/10.1172%2Fjci.insight.129348

Response: There previous studies have been carried out on either primary mature monocytes or macrophages, whereas the current study investigates  the effect of MM EVs on an immature monocyte cell line. One of these studies does provide correlating evidence that packaged components of MM EVs regulate cytokine signalling through NF-ĸB activation in the recipient cell. This is included in the revised manuscript – “Interesting, the absence of miR-16 in MM-EVs was shown to enhance NF-ĸB activation in mature primary monocytes resulting in polarization of monocytes to M2 macrophages [56]. Collectively, these studies confirm that MM EV components regulate cytokine signalling in recipient cells, and is in agreement with this study.” (line 696)

Minor points:

1)    Keep cell line, protein names consistent (include hyphens/period) e.g., THP-1, IL-6, MM1.S etc.

In the modified manuscript, THP-1, IL-6, MM.1S and MMP-9 have been edited to ensure consistency.

2)    Line 448: Is that a typo of two-way ANOVA?

They have been corrected in the revised manuscript.

3)    Fig 2C legend must include what the green arrow points to

They have been corrected in the revised manuscript.

4)    Fig 5B: (Typo) Principal component

They have been corrected in the revised manuscript.

5)    Fig 8A, B, C: X axis: typo in p.adjust

They have been corrected in the revised manuscript.

6) Fig 8D: what parameter does the scale bar represent?

Response: The scale bar represents the fold change difference within the analysis. The fold change cut-off for significance was set at 2 and this is the maximum included on the scale bar. Based on this, the inclusion of the scale bar was determined relevant to indicated enrichment and reduction of proteins but not to provide quantification of the fold change.

Comments on the Quality of English Language

Overall the manuscript is well written and easy to understand. However, a few typos need to be corrected:

1)    Keep cell line, protein names consistent (include hyphens/period) e.g., THP-1, IL-6, MM1.S etc.

They have been changed in the revised manuscript.

2)    Line 448: Is that a typo of two-way ANOVA?

They have been changed in the revised manuscript.

3)    Fig 5B: (Typo) Principal component

They have been changed in the revised manuscript.

4)    Fig 8A, B, C: X axis: typo in p.adjust

They have been changed in the revised manuscript.

Reviewer 2 Report

Comments and Suggestions for Authors

In recent years, lot of evidence has come out demonstrating the communication between cells through EVS and, in particular, their role in tumor progression in onco-hematological malignancies.

Interestingly, in this manuscript, authors investigate MM EV role in tumor growth and metastasis. In particular, they demonstrated that MM EV uptake by monocytes increased IL-6 and MMP9 production, favouring MM cells proliferation and migration.

All experiments are well conducted and have logical consequentiality, manuscript is clear and well written.

I would like the authors to clarify some small doubts about Methods:

First of all, in EV isolation they performed 35 min ultracentrifugation to wash EV pellet, isn't there a danger of something being lost since the other centrifuges for EV isolation are much longer (for example 2h and 30 min or 15 h)?

In the Nanoparticle tracking analysis method paragraph authors wrote that they performed “15 capture for 60s/captures” (line 169). Are 100 microliters of EV sample enough to make 15 videos? It is recommended to enter the dilution used for the analysis.

Authors claim to quantify EVs by flow cytometer, I believe that the instrument, as used, is just a way to visualize them. Usually, to quantify cells, and therefore also EVs, by flow cytometry, beads of known concentration should be used to apply formula and calculate sample concentrations (as reported by Laurenzana I et al. Int J Nanomedicine. 2021 May 7;16:3141-3160). With this premise, I ask how the authors can say that they have treated the cells with the amount of 25-400 EVs/cells. They should indicate the given concentration by calculating it with the NTA data or quantify it correctly by flow cytometer and insert the correct values in the paper.

Figures:

In general, for easier reading, authors should remove methods from the figure captions and leave only the result that describes them.

Fig.1: insert 200 µm value on the scale bar in the figure.

Fig. 2 is missing the title. Moreover authors should insert a representative flow cytometry histogram plot, for example an overlay between untreated and treated cells.

All supplementary figures, in the text, are indicated with “Fig. n°, suppl1”, authors may refer to them as “Fig. Sn°” as indicated in the supplementary materials.

Please, check and use the right abbreviations in the text.

Round 2

Reviewer 1 Report

Comments and Suggestions for Authors

The authors have sufficiently addressed all the reviewer comments either by redirecting to relevant material in the original manuscript or inclusion of additional points in the revised one. Overall, the manuscript is very well written and provides resources that can be useful to study how MM derived EVs may influence cells in tumor microenvironment.